# Immunity in the Progeroid Model of Cockayne Syndrome: Biomarkers of Pathological Aging

**DOI:** 10.3390/cells13050402

**Published:** 2024-02-26

**Authors:** Khouloud Zayoud, Asma Chikhaoui, Ichraf Kraoua, Anis Tebourbi, Dorra Najjar, Saker Ayari, Ines Safra, Imen Kraiem, Ilhem Turki, Samia Menif, Houda Yacoub-Youssef

**Affiliations:** 1Laboratory of Biomedical Genomics and Oncogenetics (LR16IPT05), Institut Pasteur de Tunis, Université Tunis El Manar, El Manar I, Tunis 1002, Tunisia; 2Faculty of Sciences of Bizerte, Bizerte 7021, Tunisia; 3Department of Neuropediatrics, National Institute of Neurology Mongi Ben Hamida, Tunis 1007, Tunisia; kraoua_ichraf@yahoo.fr (I.K.);; 4Orthopedic and Trauma Surgery Department, Mongi Slim Hospital, La Marsa 2070, Tunisia; tborbianis@gmail.com (A.T.); ayarisakr89@gmail.com (S.A.); 5Laboratory of Molecular and Cellular Hematology (LR16IPT07), Institut Pasteur de Tunis, Université Tunis El Manar, El Manar I, Tunis 1002, Tunisia; ines.safra@pasteur.tn (I.S.);

**Keywords:** Cockayne syndrome, progeroid syndrome, inflammaging, immunosenescence

## Abstract

Cockayne syndrome (CS) is a rare autosomal recessive disorder that affects the DNA repair process. It is a progeroid syndrome predisposing patients to accelerated aging and to increased susceptibility to respiratory infections. Here, we studied the immune status of CS patients to determine potential biomarkers associated with pathological aging. CS patients, as well as elderly and young, healthy donors, were enrolled in this study. Complete blood counts for patients and donors were assessed, immune cell subsets were analyzed using flow cytometry, and candidate cytokines were analyzed via multi-analyte ELISArray kits. In CS patients, we noticed a high percentage of lymphocytes, an increased rate of intermediate and non-classical monocytes, and a high level of pro-inflammatory cytokine IL-8. In addition, we identified an increased rate of particular subtypes of T Lymphocyte CD8+ CD28− CD27−, which are senescent T cells. Thus, an inflammatory state was found in CS patients that is similar to that observed in the elderly donors and is associated with an immunosenescence status in both groups. This could explain the CS patients’ increased susceptibility to infections, which is partly due to an aging-associated inflammation process.

## 1. Introduction

Aging is a multifactorial phenomenon that affects several organs [1]. It represents the main risk factor for many diseases, particularly cancer, neurodegenerative disorders, sarcopenia, and diabetes. The immune system is dysregulated during aging [2]. A growing number of studies suggest that the aging of the immune system contributes to the morbidity and mortality of the elderly [3,4]. Indeed, due to many age-related physiological changes in both innate and adaptive immunity, the elderly show increased susceptibility to infections such as influenza and bacterial pneumonia. For example, influenza is responsible for more than 500,000 deaths per year worldwide, two-thirds of which occur in people over 65 years old [5]. During aging, the functions of innate immune cells, such as phagocytosis, antigen uptake and presentation, migration and bactericidal activity, are diminished [6].

Several studies now indicate that the aging process is associated with a progressive increase in cellular damage [7]. Immune cells are vulnerable to endogenous DNA damage, which causes cell death or cell senescence [8]. Thus, the immune system loses its effectiveness, and therefore, it does not properly eliminate damaged cells [9,10]. Senescent cells have the capacity to secrete a myriad of factors known as the senescence-associated secretory phenotype (SASP), which includes growth factors, pro-inflammatory cytokines, chemokines, and proteases, all of which can contribute to a state of low-grade chronic inflammation, commonly known as inflammaging [4]. The production of these inflammatory mediators is regulated through transcription factors such as NF-*K*B, which is a ubiquitous transcription factor [11]. NF-*K*B signaling is one of the crucial mediators of inflammatory responses [12], and it is promoted through the secretion of SASP in an autocrine manner via cytokine receptors [13]. Furthermore, monocyte subsets, which represent the main actors involved in innate immunity, are distinguished from each other by their transcriptional signature and their functions in maintaining homeostasis [14]. Human monocytes are subdivided into three sub-groups: classical monocytes (CD14+ CD16−), non-classical monocytes (CD14− CD16+), and intermediate monocytes (CD14+ CD16+) [14]. Classical monocytes are usually involved in phagocytosis; intermediate monocytes are involved in antigen presentation and cytokine secretion, while non-classical monocytes are involved in complement and Fc-gamma-mediated phagocytosis [15]. Nevertheless, a high proportion of these subtypes is associated with a variety of aging-associated diseases with increasing severity linked mainly to the high prevalence of CD16+ intermediate and non-classical subsets [16]. Indeed, impaired transcriptional and biological responses in monocyte subsets were noted during regular aging, suggesting that aging influences the cytokine profiles of monocytes and their agonist-provoked responses [17]. 

Regarding adaptive immunity, several studies have shown age-related changes in different immune cell subtypes and functions, with a predominance of the T lymphocyte (LT) memory phenotype with a highly differentiated state in the elderly. In addition, it was shown that the differentiation of LT cells toward a senescent phenotype is typically characterized by the clonal expansion of CD8+ LT that lacks the expression of CD28, which is an important co-stimulatory molecule for LT cell activation [18,19]. The CD28 is a co-stimulatory molecule and a tumor necrosis factor (TNF) receptor which is required for the generation and maintenance of LT cell immunity by regulating B cell activation and immunoglobulin synthesis [20,21]. The low expression of the CD27 is also considered to be a biomarker of senescence [22].

Progeroid syndromes recapitulate regular aging in many aspects [23,24]. Indeed, Cockayne syndrome (CS) is a rare autosomal recessive neurodegenerative disorder that predisposes people to accelerated aging. It is mainly characterized by psychomotor retardation, cerebral atrophy, microcephaly, myelin defects, brain calcifications, mental retardation, sensorineural hearing loss, and skin photosensitivity [25]. Two main forms define this disease: (CS-A OMIM 609412) and (CS-B OMIM 609413). Although the phenotypes are similar, each form is specifically associated with mutations in either the ERCC8/CSA or ERCC6/CSB gene. CSA protein belongs to the E3 ubiquitin ligase complex required for TCR-NER DNA repair [26], while the CSB is a member of the SWI2/SNF2 ATPase family, known for its involvement in transcription, DNA separation, chromatin maintenance, and remodeling [27]. Mortality usually occurs in early adolescence, around the age of 12 [28,29,30]. CS patients are also susceptible to infections, which constitute one of the major reasons for their early death following pneumonia [31].

A study that focused on Werner syndrome, which also predisposes individuals to accelerated aging, showed that patients had low-grade inflammation characterized by high rates of inflammatory cytokines and mediators, such as C-reactive protein (CRP), IL-6, IL-8, and TNF in their blood [32,33].

Recently, we have shown that patients with CS (both the CSA and CSB forms) present a clinical heterogeneity, despite sharing the same genetic mutations [34,35]. In addition, we noted patients’ susceptibility to recurrent respiratory infections [36], which prompted us to study their immune system status. Data regarding the immune landscape in CS patients are very limited [37] given the scarcity of biological material related to these pathologies. Given that CS predisposes individuals to accelerated aging, we compared the phenotypes of immune cells, as well as key players of immunity in CS patients, to those of elderly donors and healthy young donors.

## 2. Materials and Methods

### 2.1. Patients and Healthy Donors

After obtaining written informed consent from the patients’ tutors (as CS patients were minors) and from the other participants (elderly and healthy donors), blood samples and clinical data were collected. This study was carried out in accordance with the Declaration of Helsinki Principles and approved by the Biomedical Ethics Committee of the Pasteur Institute of Tunisia. The Ethical Approval Number is 2017/31/I/LR16IPT05/V2.

### 2.2. Complete Blood Counts (CBC)

Complete blood counts (CBC) of CS patients, healthy elderly and healthy young donors were performed. A total of 413 CBC were analyzed and divided according to the following age groups of the donors (Grp 1: 40 cases [1–6 years]; Grp 2: 36 cases [7–12 years]; Grp 3: 24 cases [13–17 years]; Grp 4: 49 cases [18–25 years]; Grp 5: 73 cases [26–40 years]; Grp 6: 66 cases [41–60 years]; Grp 7: 57 cases [61–75 years]; Grp 8: 52 cases [76–95 years] and Grp 9: 14 CS patients [7–12 years]). 

### 2.3. Cytokines Measurements by Multi-Analyte ELISArray Kit

A total of six samples were used from genetically confirmed CS patients aged 5–8 years. In addition, we used 6 samples of serum from healthy donors, which we divided into two groups according to their ages: young healthy donors aged 7–12 years and elderly donors aged 71–85 years. Twelve cytokines and chemokines including IL-1α, IL-1b, IL-2, IL-4, IL-5, IL-6, IL-8, IL-10, IL-12, IL-13, IL-17 and GM-CSF were measured simultaneously in the serum of patients and donors using a Multi-Analyte ELISArray kit (MEH-006A; REF: 336161 QIAGEN, Germantown, MD, USA) according to the manufacturer’s instructions. The absorbance was measured after 30 min of incubation at 450 nm and cytokine/chemokine levels were determined after comparing them to the negative control (assay buffer) and to the positive control (cocktail containing the standards of 12 cytokines/chemokines). The detection of a cytokine/chemokine was based on an absorbance value above the negative control. The mean absorbance was calculated based on duplicates for each sample and all standards. The concentrations were calculated using standard curves. Cytokine/chemokine levels were measured in pg/mL, according to previous protocols [38,39].

### 2.4. Immune Cell Phenotyping Using Flow Cytometry

Immunolabelling was performed for PBMC (Peripheral Blood Monocyte cells) isolated from the blood of four CS patients, and from healthy donors that we divided into four groups according to their ages (Grp1: 4 healthy donors [18–29 years], Grp2: 7 healthy donors [30–45 years], Grp3: 12 healthy donors [46–65 years] and Grp4: 4 CS patients). The immunolabelling was performed for the following surface markers: LT cells (CD3 [ref:555342-BD], CD4 [ref:555348-BD], CD8 [ref:555634-BD], CD28 [ref:561368-BD], CD27 [ref:557330-BD]); LB cells (CD19 [ref:555412-BD], CD20 [ref:555624-BD]); NK cells (CD16 [ref:561308-BD], CD56 [ref:560916-BD]) and monocytes (CD14 [ref:561709-BD], CD16 [ref:561308-BD]). Elderly people aged over 70 years, with diabetes, were excluded from this study. The gating strategy has been illustrated in the Appendix A with a minimal amount of 10.000 events for each analyzed sample. 

### 2.5. Statistical Analyses

Statistical analysis was performed using the GraphPad Prism 10. The distribution of variables was checked for normality with the Kolmogorov–Smirnov test. The values of biochemical parameters in the different groups were compared with non-parametric Tukey and Kruskal–Wallis tests for continuous variables. The levels of statistical significance were considered as follows: * (*p* ≤ 0.05); ** (*p* ≤ 0.01); *** (*p* ≤ 0.001); **** (*p* ≤ 0.0001).

## 3. Results

### 3.1. Variations in Immune Parameters in CS Patients, and Comparison with Those of Elderly and Young Healthy Donors

Values of complete blood count (CBC) parameters: red blood cells (RBCs), hemoglobin (HBG), white blood cells (WBCs), lymphocytes, monocytes and neutrophils for CS patients (N = 14); in total, 8 groups of healthy donors were assessed.

#### 3.1.1. Increased Red Blood Cell (RBC) Count in CS Patients and the Elderly

Statistical analysis showed that the red blood cells (RBCs) were significantly decreased in the elderly (76–95 years: 3.812 ± 0.72) compared to the healthy young donors (1–6 years: 4.721 ± 0.55; 7–12 years: 4.775 ± 0.518) (*p* ≤ 0.0001).

#### 3.1.2. Low Levels of Hemoglobin (HBG) in CS Patients and Elderly

Statistical analysis revealed that hemoglobin (HBG) was significantly decreased in the elderly (76–95 years: 10.97 ± 2.17) when compared to the healthy young donors (7–12 years) (12.67 ± 1.42) (*p* = 0.0044). Similarly, the CS group (10.64 ± 2.088) exhibited a significant decrease in HBG when compared to the healthy young donors (7–12 years; *p* = 0.04) (Figure 1).

#### 3.1.3. Increased Level of White Blood Cells (WBCs) in CS Patients and Elderly

We have noted significant differences between healthy young donors (7–12 years) and elderly donors (76–95 years). Statistical analysis showed that WBCs were significantly increased in the elderly (76–95 years: 10.36 ± 3.27) compared to the healthy young donors (7–12 years: 6.731 ± 2.11) (*p* ≤ 0.0001). Similarly, the CS patients’ group (10.37 ± 3.674) showed a significant increase in WBCs compared to the same young donors’ group (7–12 years) (*p* = 0.0167) (Figure 1).

#### 3.1.4. Decreased Rate of Lymphocytes Observed in The Elderly Group

During regular aging, the lymphocytes decrease continuously. The statistical analysis revealed that the percentage of lymphocytes decreases significantly in the elderly (76–95 years) compared to the healthy young donors (1–6 years: 48.81 ± 11.09; 7–12 years: 40.25 ± 10.26; 13–17 years: 38.11 ± 13.44 and 18–25 years: 36.45 ± 11.40) (*p* ≤ 0.0001). Interestingly, statistical analysis showed a significant increase in the CS patients’ group compared to the healthy donors (26–40 years: *p* = 0.014) and to the elderly (41–60 years: *p* = 0.0015; 61–75 years: *p* = 0.001; 76–95 years:14.69 ± 9.5; *p* ≤ 0.0001) (Figure 1).

#### 3.1.5. Increased Rate of Monocytes in CS Patients

Statistical analysis showed that monocytes were significantly decreased in the group aged (61–75 years: 6.031 ± 1.96 and 76–95 years: 5.322 ± 2.27) compared to the healthy young donors (1–6 years: 9.961 ± 3.135) (*p* ≤ 0.0001). While the CS patients’ group showed an increased percentage of monocytes compared to healthy donors (13–17 years: *p* = 0.0005; 18–25 years: *p* = 0.0048; 26–40 years: *p* = 0.0005) and to the elderly (41–60 years: *p* = 0.0006; 61–75 years: *p* ≤ 0.0001; 76–95 years; *p* ≤ 0.0001) (Figure 1).

#### 3.1.6. Increased Rate of Neutrophils in Elderly

In this work, statistical analysis showed that neutrophils increase with age as in the elderly (76–95 years) compared to the healthy young donors’ groups (1–6 years: 45.28 ± 6.981; *p* = 0.0023 and 7–12 years: 45.61 ± 11.49; *p* = 0.0032), whereas the CS patients’ group (41.64 ± 12.59) showed a significant decrease for these cells when compared to the elderly (76–95 years: 73.61 ± 18.31; *p* = 0.0067) (Figure 1).

Based on these results, we noticed that the CS patients showed a similar pattern for the three studied parameters (white blood cells, red blood cells and hemoglobin) as for the group of elderly (61–75 years; 76–95 years). Regarding the other studied parameters (lymphocytes, monocytes and neutrophils), CS patients were closer to the healthy young donors’ groups (1–6 years; and 7–12 years). 

### 3.2. Increased Cytokine Levels in CS Patients

Serum from CS patients and elderly were analyzed using multi-analyte ELISArray (Qiagen kits), which assays 12 pro and anti-inflammatory cytokines in the same time (IL-1a, IL-1b, IL-2, IL-4, IL-5, IL-6, IL-8, IL-10, IL-12, IL-17a, IL-13 and GM-CSF. We only report cytokines that were above the detection threshold in the different groups. Results showed that the anti-inflammatory cytokines IL-4, IL-5, IL-10 and IL-13 slightly increased in CS patients compared to healthy donors. It is not statistically significant; this could be explained by the low number of CS patients who are very difficult to recruit. [IL-4: (7–12 years: 33.08 ± 3.7; 71–85 years: 34.08 ± 7.36; CS: 76.87 ± 77.9; *p* > 0.99); IL-5: (7–12 years: 18 ± 14; 71–85 years: 18.27 ± 3; CS: 68.27 ± 77.06; *p* > 0.99); IL-10 (7–12 years: 33.06 ± 3.1; 71–85 years: 54.72 ± 8.9; CS: 102.8 ± 95.51; *p* > 0.99); IL-12 (7–12 years: 17.63 ± 3.7; 71–85 years: 15.14 ± 3.3; CS: 14.7 ± 3.101; *p* > 0.99); IL-13 (7–12 years: 19.42 ± 0.59; 71–85 years: 19.42 ± 6.2; CS: 70.36 ± 72.33; *p* > 0.99)]. Interestingly, the pro-inflammatory cytokine IL-8 was significantly increased in CS patients (313.2 ± 370) compared to the healthy young donors (27.14 ± 6.216) and to the elderly (45.43 ± 21.21; *p* = 0.0218) (Figure 2).

### 3.3. Variations in Immune Cell Subsets in CS Patients Compared to Healthy Young Donors and the Elderly

#### 3.3.1. Decreased Rate of CD4+ LT in CS Patients

The percentage of CD4+ LT cells was significantly decreased in CS patients (24.43 ± 17.4) compared to both healthy young (18–29 years; 50.43 ± 9.16; *p* = 0.03), (30–45 years; 49.17 ± 6.251; *p* = 0.0207) and elderly groups (46–65 years; 48.38 ± 14.6; *p* = 0.01) (Figure 3a). The rate of CD4+ CD28+ CD27+ LT cells was similar to that in the healthy elderly group (46–65 years) (Figure 3b). However, the percentage of these lymphocytes was lower in healthy young donors (18–29 years). 

#### 3.3.2. Increased Rate of CD8+ LT Senescent Phenotype in CS Patients

Thereby, age-associated changes in adaptive immunity, specifically in the LT population, seem to be pronounced in CD8+ LT more than in CD4+ LT. In fact, the CD8+ CD28− CD27− LT subtype was increased in CS patients (30.48 ± 5.19) compared to healthy donors (18–29 years: 6.86 ± 9.02; *p* = 0.0023 and to 30–45 years: 14.30 ± 11.1; *p* = 0.024) (Figure 3f), while the CD8+ CD28+ CD27+ LT subtype was significantly decreased in CS patients (45.6 ± 5.23) compared to healthy young donors (18–29 years: 79.3 ± 15.08; *p* = 0.0081), and to elderly group (46–65 years: 48.48 ± 12.45; *p* = 0.029) (Figure 3e).

#### 3.3.3. Constant LB and NK Cell Percentages in CS Patients and in Healthy Donors

No statistical differences were found regarding the percentages of CD19+ CD20+ LB cells between CS patients and both healthy elderly and healthy young donors. No significant differences were found regarding the NK cell percentages, between all three groups as well (Figure 4).

#### 3.3.4. Increased Rate of Intermediate and Non-Classical Monocyte Subsets in CS Patients

The percentages of the classical (C) monocytes (CD14++ CD16−) showed a significant decrease in the CS patients’ group (37.08 ± 13.95; *p* < 0.001) and in the elderly (46–65 years: 63.69 ± 11.49; *p* = 0.0004) compared to the healthy donors’ group (18–29 years: 78.6 ± 7.03) (Figure 5a). The number of the non-classical (NC) monocytes (CD14+ CD16++) was statistically higher in the CS patients’ group (26.28 ± 14.08) than in the healthy donor group (30–45 years: 2.44 ± 1.76; *p* = 0.007) (Figure 5c). Finally, the subset of intermediate (I) monocytes (CD14++, CD16+) significantly increased in the CS patients’ group (25.53 ± 3.7; *p* < 0.0001) and in the elderly (46–65 years: 15.63 ± 5.94; *p* ≤ 0.0001) compared to the healthy donors’ group (18–29 years: 6.39 ± 2.3) (Figure 5b). Thus, monocyte subsets (intermediate monocytes and non-classical monocytes) were significantly higher in CS patients compared to healthy young donors.

## 4. Discussion

Aging is characterized by a progressive decline in the immune system functions. The study of immunity in progeroid models predisposing to pathological and accelerated aging will help to determine the main biomarkers of aging to prevent age-related diseases. This is the case of our study model of Cockayne syndrome (CS).

Furthermore, given the high susceptibility of patients with CS to infections, we believe that immunity is necessarily involved in this vulnerability, which is also observed in the elderly. Hence the importance and originality of our study. In addition, since CS is an extremely rare disease, the immunity responses in CS patients have never been explored. In this work, we studied the hematological parameters as well as the inflammatory cytokines and immune cell phenotypes to map the immune landscape in CS, on the one hand, and to identify the main biomarkers involved in accelerated aging, on the other hand.

### 4.1. CBC Parameters in Healthy Donors and in CS Patients

In this work, we found that red blood cell percentages and hemoglobin levels decreased in CS patients and in elderly donors compared to young donors. These two CBC parameters are closely associated and correlated. Previous work has shown that the decline in red blood cells and hemoglobin is accentuated with age [40,41,42], which is often associated with anemia. This anemia in the elderly is a consequence of poor absorption of iron, vitamin B12, and folate as well as gastrointestinal bleeding [43,44].

Analysis of white blood cell (WBC) percentages showed a significant increase in CS patients and in the elderly compared to healthy young donors. Previous studies have shown that with age, the percentage of WBCs increases significantly and that it is associated with an increased mortality risk in the elderly [45,46]. A high number of WBCs is associated with frailty in the elderly, and with biological vulnerability to multiple stress factors [47]. Here we showed that CS patients present an increase in WBCs, as well as in the elderly donors, which exposes them to multiple pathological risks.

Detailing the composition of WBCs, we first examined the percentages of lymphocytes and monocytes which decrease significantly in the elderly compared to young healthy donors, whereas, the percentage of neutrophils increases with age. Interestingly, the percentages of lymphocytes, monocytes and neutrophils in CS patients were preserved and similar to those in young healthy donors at similar ages. Previous studies have shown that neutrophils and monocytes increase significantly with age while lymphocytes’ number decreases significantly [48]. Indeed, with age, the number of circulating lymphocytes decreases as well as lymphocyte functions [49]. It has been attributed to a thymus atrophy [50]. However, neutrophils’ number increases during aging and their phenotype changes to CD62LlowCXCR4+ neutrophils with high microbicidal activity and phagocytic capacity [51]. Myeloid innate immune cells (neutrophils, monocytes and dendritic cells), whose number and function are altered during aging, represent the main source of inflammatory mediators, responsible for a chronic low-grade inflammation known as “inflammaging” [52]. 

Based on all these data, we showed that CS patients were similar to young donors (of the same age) for three parameters: lymphocytes, monocytes and neutrophils, while they were similar to the elderly for red blood cells, hemoglobin and WBC parameters. 

### 4.2. Cytokines Level in Healthy Donors and in CS Patients

In this study, 12 pro- and anti-inflammatory cytokines were investigated in young and elderly healthy donors, as well as in CS patients. A significantly high level of pro-inflammatory cytokine, IL-8, was found in CS patients compared to both healthy elderly and young donors. Four anti-inflammatory cytokines, IL-4, IL-5, IL-13 and IL-10, were also slightly increased in CS patients compared to both healthy donor groups. 

High serum rates of IL-1α, IL-6, IL-8 and IL-12 were previously associated with aging [53,54,55,56,57]. Moreover, in Werner patients, another progeroid syndrome, a Th2-like secretory phenotype has been found with high serum levels of IL-4 and IL-10 [32]. Furthermore, increased levels of IL-8 were associated with a high risk of mortality in elderly [54]. The IL-8, is mainly regulated by the transcription factor NF-*K*B, it is a chemokine secreted by monocytes/macrophages and plays a key role in neutrophils’ recruitment and activation [21]. A high level of IL-8 disrupts neutrophils’ migration [58]. Increased level of IL-8 was also proposed as a biomarker of age-associated-chronic-inflammation in the lungs [59]. Its involvement in many inflammatory diseases, such as chronic lung diseases, inflammatory bowel diseases and certain autoimmune diseases has been previously described [57]. We demonstrate here a significant increase in IL-8 in the progeroid CS model which is associated with an accelerated and pathological aging and would be partly responsible for age-related pathologies.

### 4.3. Immune Cell Phenotypes in CS and in Elderly

We showed that with age, there are changes in adaptive immunity, particularly in the LT population, which appear to be more pronounced in the CD8+ LT, than in the CD4+ LT subsets. Indeed, the CD8+ CD28− CD27− LT subset, which represents the senescent phenotype of LT, is higher in CS patients and in elderly donors, than in young healthy donors. While the CD8+ CD28+ CD27+ LT subset, which represents the active form, is significantly reduced in CS patients and in elderly donors compared to young healthy donors. Our results are in accordance with previous studies showing that the loss of CD28 surface marker expression, is a feature of senescent LT [19,60] and that the loss of this marker is more pronounced in CD8+ LT than in CD4+ LT [18,61]. Indeed, the loss of CD28 expression is associated with telomerase inactivation, and as a result, CD8+ CD28− LT cells have shorter telomeres than CD8+ CD28+ LT cells [62,63,64]. Other studies have shown that CD27 expression can identify the most senescent CD8+ CD28− LT, indicating that CD8+ CD28− CD27− LT cells have the shortest telomeres [22]. Interestingly, it has been reported that senescent LT cells predispose to lung infections in the elderly [65,66]. This is thought to be the reason for the susceptibility of CS patients and elderly to multiple respiratory infections such as pneumonia [31]. 

Moreover, it was shown that CD8+ LT senescent cells express surface receptors that are associated with NK cells, such as the NKG2D, NKG2C, NKG2A, and killer immunoglobulin-like receptor (KIR) families compared to undifferentiated, non-senescent CD28+ CD27+ CD8+ LT [67,68,69,70]. This suggests that rather than being dysfunctional, these cells acquire an alternative functional profile as they differentiate. This hypothesis is supported by the observation that these T-senescent cells express DAP12, an adaptor molecule of NK cytotoxicity, and are able to kill tumor cells in an MHCI-independent manner [69]. Indeed, a recent work has demonstrated that the switch from TCR to NKR expression in CD8+ LT senescent cells is regulated by stress proteins known as sestrins in both human and mice [69]. Of note, CD4+ LT-senescent cells also express NKRs, suggesting that they may also mediate effector functions through these receptors [69]. Therefore, LT-senescent cells may mediate NKR-dependent cytotoxicity, independently of their antigen specificity. 

Furthermore, we showed in this study that the rates of monocyte subtypes such as intermediate monocytes (CD14++ CD16+) and non-classical monocytes (CD14+ CD16++) were higher in CS patients and in healthy elderly donors compared to young healthy donors. Our results were consistent with a previous study that found the same pattern in the elderly [71]. The authors attributed this to age-associated chronic inflammation, which is mediated mainly by non-classical monocytes, intermediate monocytes and increased levels of IL-8, and probably induced by the NF-*Κ*B pathway.

In this study, we showed that CS patients share the same immune parameters with elderly donors, as for decreased red blood cells and hemoglobin, as well as an increased rate of WBCs and senescent CD8+ LT. However, they present some particularities, such as the significant increase in IL-8 and non-classical monocytes, which reflect a chronic inflammatory status in CS patients known as an inflammaging state. 

Now, further studies are needed to deeply characterize the CD8+ LT subtypes as well as the monocyte sub-populations, via RNA-sequencing following cell sorting, or alternatively, performing single-cell analysis, which would help to precisely define the molecular biomarkers involved in the accelerated aging process. However, this remains a significant challenge, particularly in the study of rare diseases such as the case of Cockayne syndrome, which is an interesting model for understanding the mechanisms of pathological aging.

## 5. Conclusions

Based on these results, we showed a particular immune landscape in CS patients, which shares certain parameters with the phenotype of regular aging and would be partly responsible for accelerated aging. However, some parameters are specific to them, which can explain their high susceptibility to infections.

## Figures and Tables

**Figure 1 cells-13-00402-f001:**
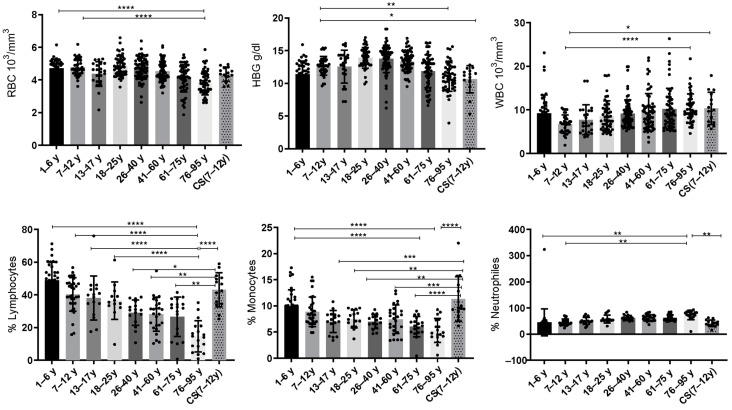
Box plots of CBC values in 14 CS patients aged (7–12 years) and in different groups of healthy donors (N = 40 [1–6 years]; N = 36 [7–12 years]; N = 24 [13–17 years]; N = 49 [18–25 years]; N = 73 [26–40 years]; N = 66 [41–60 years]; N = 57 [61–75 years]; and N = 52 [76–95 years]). The results are presented as individual values with mean ± SD. The difference in mean values between the different groups was determined using the Tukey test. N = number of subjects. Asterisks indicate statistical significance: * (*p* ≤ 0.05); ** (*p* ≤ 0.01); *** (*p* ≤ 0.001); **** (*p* ≤ 0.0001).

**Figure 2 cells-13-00402-f002:**
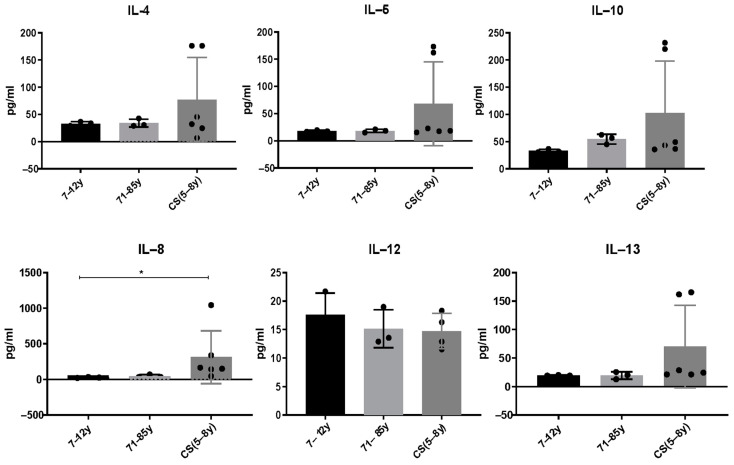
Cytokine levels in CS patients (N = 6 cases, 5 to 8 years) and in healthy donors (N = 3 [7–12 years]; N = 3 [71–85 years]). Sera were collected and analyzed by a Multi-Analyte ELISArray kit for 12 cytokines. Only cytokines/chemokines above the absorbance value of the negative control were considered. Results are shown based on duplicates for each sample. The difference in mean values between the different groups was determined using the Kruskall–Wallis test. N = number of subjects. Asterisks indicate statistical significance: * (*p* ≤ 0.05).

**Figure 3 cells-13-00402-f003:**
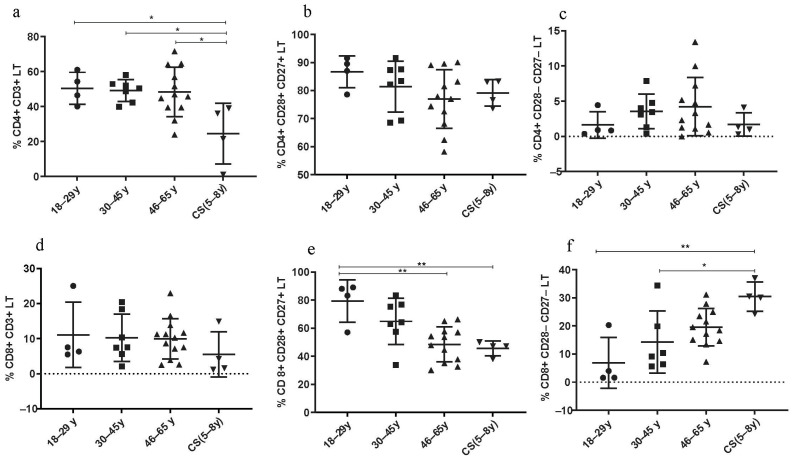
Lymphocyte subset percentages in CS patients (N = 4 [5–8 years]) and healthy donors (N = 4 [18–29 years]; N = 7 [30–45 years]; N = 12 [46–65 years]). (**a**) CD4+ CD3+ LT; (**b**) CD4+ CD28+ CD27+ LT; (**c**) CD4+ CD28− CD27− LT; (**d**) CD8+ CD3+ LT; (**e**) CD8+ CD28+ CD27+ LT; (**f**) CD8+ CD28− CD27− LT. Percentages of the different TCD4 and TCD8 subsets were analyzed by flow cytometry. Statistical analysis was performed using the Kruskal–Wallis test. Lines represent the medians, and each dot represents a donor. N = number of subjects. Asterisks indicate statistical significance * *p* < 0.05, ** *p* < 0.01. A dashed horizontal line at 0% serves as the cutoff value, indicating the threshold for defining the groups used for subsequent statistical analysis.

**Figure 4 cells-13-00402-f004:**
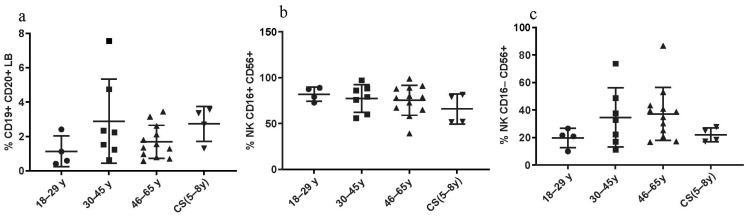
LB and NK cell phenotyping: Percentages in CS patients (N = 4 [5–8 years]) and in three groups of healthy donors (N = 4 [18–29 years]; N = 7 [30–45 years]; N = 12 [46–65 years]). (**a**) CD19+ CD20+ LB; (**b**,**c**) NK subsets, assessed by flow cytometry. Statistical analysis wasthe performed using the Kruskal–Wallis test. Lines represent the medians, and each dot represents a donor. N = number of subjects.

**Figure 5 cells-13-00402-f005:**
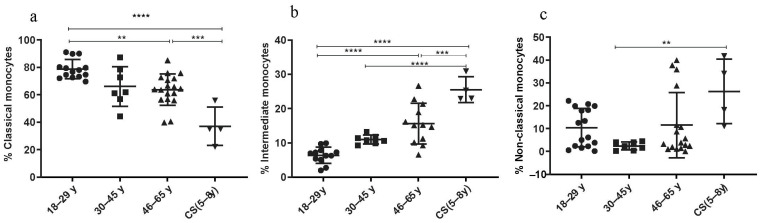
Monocyte subsets phenotyping. Percentages of monocyte subsets in CS patients (N = 4 [5−8 years]) and in three healthy donor groups (N = 4 [18−29 years]; N= 7 [30−45 years]; N = 12 [46−65 years]). (**a**) Classical monocytes; (**b**) intermediate monocytes; (**c**) non-classical monocytes. Percentages of studied cells analyzed by flow cytometry. Statistical analysis was performed using the Kruskal–Wallis test. N = number of subjects. Asterisks indicate statistical significance: ** (*p* ≤ 0.01); *** (*p* ≤ 0.001); **** (*p* ≤ 0.0001).

## Data Availability

All processed data have been provided in the manuscript. The corresponding author upon reasonable request could provide raw data, generated for this study.

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
