# Peer review of "Immunity in the Progeroid Model of Cockayne Syndrome: Biomarkers of Pathological Aging"

_cells, 2024, doi:10.3390/cells13050402_

Round 1
Reviewer 1 Report
Comments and Suggestions for Authors
The introduction is very well detailed and written. However, as a non-expert in the Cockayne syndrome, I believe that authors should add more information about the disease (Forms (CSA/CSB), mutations, life span). I found this paper interesting: Wilson, B., Stark, Z., Sutton, R. et al. The Cockayne Syndrome Natural History (CoSyNH) study: clinical findings in 102 individuals and recommendations for care. Genet Med 18, 483–493 (2016). https://doi.org/10.1038/gim.2015.110
Regarding the results part and the figures, authors have a lot of work to do: scientifically and structurally.
First, regarding the structure:
- the result titles should summarize the result found and not just described what has been done.
- Figure legends are missing everywhere. Authors should describe the experiment in the legend and add which statistical test was done and how many patients per group.
- For histogram, adding the dots for each patient would be good, as it shows the distribution.
Secondly, regarding their findings:
- Authors should indicate the age of the CS patients compare this group to the corresponding healthy group to show differences as well (everywhere).
- “Interestingly, the CS patients exhibited nearly the same pattern for the examined parameter as in elderly group (61-75 years) ”. Not statistically significant. Authors cannot conclude. And “by eye” it seems closer to 41-60 than 61-75.
- For Lymphocyte and Monocyte counts, there is an increase but comparison between CS and young must be done. Same for Neutrophils. Not sure there is a difference in neutrophils with young healthy donors.
- Because there are only 4 patients, it is difficult to interpret the cytokine data. Authors must show all the cytokine analyzed. Having patients’ samples must be difficult to have, but results must be clear to be trust.
- The gating strategy is not a result but a method and could be placed in supplemental. Are 10,000 events enough per samples? The % are not clear to understand. Is it % of population to total population (ex: %CD4+CD3+LT/all CD3+)? Figure 4c is not significant, authors cannot conclude that there is a decrease! A major problem is that there is no aged-control group (young healthy) to compare CS. The result for CD8 T cells is very interesting. The monocyte result is also interesting, but authors should try to find if this population behave differently in CS (RNAseq on sorted population)?
Globally, these results are a good start, but now with the area of single-cell the paper miss deeper characterization of CD8 and mono-mac populations. Authors could try to sort out these populations are do RNAseq or directly go through single cell.
Author Response
Reviewer 1 Comments
Reviewer 1
1) The introduction is very well detailed and written. However, as a non-expert in the Cockayne syndrome, I believe that authors should add more information about the disease (Forms (CSA/CSB), mutations, life span).
I found this paper interesting: Wilson, B., Stark, Z., Sutton, R. et al. The Cockayne Syndrome Natural History (CoSyNH) study: clinical findings in 102 individuals and recommendations for care. Genet Med 18, 483–493 (2016). https://doi.org/10.1038/gim.2015.110
R1. We appreciate the reviewer's insightful feedback on the introduction and their suggestion to provide more information about Cockayne syndrome. We have added further information that expands on these points, Line 81-87.
Regarding the results part and the figures, authors have a lot of work to do: scientifically and structurally.
First, regarding the structure:
2a) The result titles should summarize the result found and not just described what has been done.
2b) Figure legends are missing everywhere. Authors should describe the experiment in the legend and add which statistical test was done and how many patients per group.
R2. We would like to thank the reviewer for his comment. Titles and legends have been modified accordingly.
3) For histogram, adding the dots for each patient would be good, as it shows the distribution.
R3. We have made the requested modification in Figure 1 and 2.
Secondly, regarding their findings:
4) Authors should indicate the age of the CS patients compare this group to the corresponding healthy group to show differences as well (everywhere).
R4. We sincerely appreciate the constructive feedback provided by the reviewer. We have indicated the age of CS patients which ranged from 7 to 12 years in CBC analysis and 5 to 8 years in flow cytometry analysis. The comparison to the corresponding healthy group has also been added.
5) “Interestingly, the CS patients exhibited nearly the same pattern for the examined parameter as in elderly group (61-75 years) ”. Not statistically significant. Authors cannot conclude. And “by eye” it seems closer to 41-60 than 61-75.
R5. We modified as follow: The CS patients seems to exhibit nearly the same pattern for the examined parameter as in elderly group (41-60years) (Figure 1). Line 156-157.
6) For Lymphocyte and Monocyte counts, there is an increase but comparison between CS and young must be done. Same for Neutrophils. Not sure there is a difference in neutrophils with young healthy donors.
R6. For lymphocytes (line 176-178) and monocytes (183-185), comparison between CS and healthy young donors were done and added to the figures. For neutrophils no difference between Cs and healthy young donors was noted.
7) Because there are only 4 patients, it is difficult to interpret the cytokine data. Authors must show all the cytokine analyzed. Having patients’ samples must be difficult to have, but results must be clear to be trust.
R7. We fully understand and acknowledge the reviewer concern about the difficulty in interpreting cytokine data with only four patients. We followed the same analysis strategy as used in previous work carried out on small cohorts, as outlined in the “method” section. However, additional validations are required, which we were unable to do due to lack of available samples. The Optical Density at 450 reads between 0 and 2.5 was considered to be within the linear range for all of the analytes as stated by the manufacturer and therefore only the cytokines within this range were maintained.
8) The gating strategy is not a result but a method and could be placed in supplemental. Are 10,000 events enough per samples? The % are not clear to understand. Is it % of population to total population (ex: %CD4+CD3+LT/all CD3+)?
R8. We deleted the gating strategy from the results section. The number of acquired events affects the precision and sensitivity of the assay. In general for flow cytometric immunophenotyping, at least 200,000 events are needed to characterize a population with adequate precision within peripheral blood as it have been stated by Andrea Illingworth et al 2020, however, we use an inhouse optimized protocol for patients with rare samples where the number of isolated cells is low. So 10.000 event were used for each sample (line 138-140).
The % of TCD3+ represents the total population of T lymphocytes (according to gating). Then the % of CD4+CD3+LT is the subpopulation extracted from the TCD3+ population that we previously delimited. This strategy was previously used in different model of studies (Nina Worel et al 2023) (Qiong Wang et al 2020).
9a) Figure 4c is not significant, authors cannot conclude that there is a decrease! A major problem is that there is no aged-control group (young healthy) to compare CS.
R9a.Regarding Figure 4c, we acknowledge your concern about the lack of statistical significance. Therefore, we deleted the information.
9b) The result for CD8 T cells is very interesting. The monocyte result is also interesting, but authors should try to find if this population behave differently in CS (RNAseq on sorted population)?Globally, these results are a good start, but now with the area of single-cell the paper miss deeper characterization of CD8 and mono-mac populations. Authors could try to sort out these populations are do RNAseq or directly go through single cell.
R9b.We would like to thank the reviewer for this important suggestion. Indeed, we wish to carry out cell sorting of these subpopulations of LTCD8 and monocytes and study it by RNAseq, which are techniques available in several research institutions in Europe and elsewhere. Unfortunately, in our country we do not have single cell or cell sorter technology. The only possibility of doing this will be in another country. However, we faced two problems: the transfer of biological material (for logistical reasons) on the one hand, and the little biological material which after freezing/thawing would not be in sufficient quantity to effectively start RNAseq . We really hope to do this in the future as part of international collaborations. We underlined it in the perspectives of this paper to emphasize the importance of going further in the investigation of these two groups of immune cells. Line 389-395.

Reviewer 2 Report
Comments and Suggestions for Authors
Authors have done this study on Cockayne Syndrome (CS), which is a rare autosomal recessive disorder impacting the DNA repair process, leading to progeroid syndrome and heightened susceptibility to respiratory infections. This study aimed to assess the immune status of CS patients, identifying potential biomarkers linked to pathological aging. CS patients, along with young and elderly healthy donors, participated in the study. Comprehensive blood counts, flow cytometry for immune cell subsets, and analysis of candidate cytokines using multi-analyte ELISArray kits were conducted. CS patients exhibited elevated lymphocyte percentages, increased intermediate and non-classical monocytes, and higher levels of the pro-inflammatory cytokine IL-8. Additionally, a specific subset of senescent T cells (Lymphocytes T CD8+ CD28- CD27-) showed an increased rate. The inflammatory state observed in CS patients resembled that of the elderly, indicating an immuno-senescence status in both groups. This inflammatory environment may contribute to the heightened susceptibility to infections in CS patients, reminiscent of processes associated with aging.
Authors have designed and answered the raised questions. There are some minor corrections needs to be done before the study can be accepted for publication.
1. In line 69, authors have used first time abbreviation LT, please provide full form of any abbreviation, when it is used first time in the text.
2. Line 81, 82 and 83 “The CS patients are also susceptible to infections which constitute one of the major reasons for their early death following pneumonia” a reference is missing.
3. Please provide the meaning of *, **, **, and **** in the Figure1 ligand too for the easy understanding of readers.
4. In Figure 2, IL-8 showing the significant difference in CS than in the young healthy and elderly healthy, but the similar trend can be seen in the IL-4, IL-5, IL-10, and IL-12 and opposite trend is seen in case of IL- 13. Please provide the p-values so that the readers can understand that how far they were from being significant.
5. In Figure 3, please put c and d with the respective figures.
6. In the Line 235 “There was no difference in the rates of CD19+ CD20+ LB cells in both healthy donors and CS group”. I don’t understand rates here.
7. Figure 5c has an unnecessary number 1 on it, please remove it.
8. In Figure 6a, there is an unnecessary number 1, Figure 6b and Figure 6c are not cropped correctly, please correct them.
Author Response
Reviewer 2 Comments
Reviewer 2
Authors have done this study on Cockayne Syndrome (CS), which is a rare autosomal recessive disorder impacting the DNA repair process, leading to progeroid syndrome and heightened susceptibility to respiratory infections. This study aimed to assess the immune status of CS patients, identifying potential biomarkers linked to pathological aging. CS patients, along with young and elderly healthy donors, participated in the study. Comprehensive blood counts, flow cytometry for immune cell subsets, and analysis of candidate cytokines using multi-analyte ELISArray kits were conducted. CS patients exhibited elevated lymphocyte percentages, increased intermediate and non-classical monocytes, and higher levels of the pro-inflammatory cytokine IL-8. Additionally, a specific subset of senescent T cells (Lymphocytes T CD8+ CD28- CD27-) showed an increased rate. The inflammatory state observed in CS patients resembled that of the elderly, indicating an immuno-senescence status in both groups. This inflammatory environment may contribute to the heightened susceptibility to infections in CS patients, reminiscent of processes associated with aging.
Authors have designed and answered the raised questions. There are some minor corrections needs to be done before the study can be accepted for publication.
- In line 69, authors have used first time abbreviation LT, please provide full form of any abbreviation, when it is used first time in the text.
R1. Appropriate full form of Lymphocyte T has been added to line 69.
- Line 81, 82 and 83 “The CS patients are also susceptible to infections which constitute one of the major reasons for their early death following pneumonia” a reference is missing.
R2. We would like to thank the reviewer for pointing this out. The missing references have been added. Line 89.
- Please provide the meaning of *, **, **, and **** in the Figure1 ligand too for the easy understanding of readers.
R3. We included a note in the figure legend to explain the meaning of the asterisks.
- In Figure 2, IL-8 showing the significant difference in CS than in the young healthy and elderly healthy, but the similar trend can be seen in the IL-4, IL-5, IL-10, and IL-12 and opposite trend is seen in case of IL- 13. Please provide the p-values so that the readers can understand that how far they were from being significant.
R4. We added P-value for the different studied cytokines (Line 211-215).
- In Figure 3, please put c and d with the respective figures.
R5. We thank the reviewer for pointing this out. We fixed it as requested.
- In the Line 235 “There was no difference in the rates of CD19+ CD20+ LB cells in both healthy donors and CS group”. I don’t understand rates here.
The sentences from (line 251-254) have been modified to better explain that there was no significant difference within the LB and NK populations between the three studied groups: CS, healthy elderly and healthy young donors.
- Figure 5c has an unnecessary number 1 on it, please remove it.
It was corrected.
- In Figure 6a, there is an unnecessary number 1, Figure 6b and Figure 6c are not cropped correctly, please correct them.
We revised the figure 6 and correct it as requested.

Round 2
Reviewer 1 Report
Comments and Suggestions for Authors
The authors have reinforced the manuscript as suggested in the first reviewing. The statistics data accompanied by the modification of the histogram strongly reinforce the manuscript and the conclusions. This shows the amount of work behind the data.
I still have a few comments that need to be modified before publications:
1) 'The CS patients seems to exhibit nearly the same pattern for the examined parameter as in elderly group (41-60years (Figure 1)". Now that we have a better overview, I don't think authors can conclude anything as it is not significant. I don't think it is necessary to conclude something here.
2) "Results showed that the anti-inflammatory cytokines IL-4, IL-5, IL-10 and IL-13 slightly increase in CS patients compared to heathy donors". Authors should emphasize here that regarding the low number of patients, it is not statistically significant but there is a trend. Authors can also re-emphasize how difficult it is to obtain patients. This is completely fine.
3) "Lymphocyte T" should be written "T lymphocyte".
4) Some images are blurry.
5) References to all product used must be indicated in the protocol. Antibodies used for flow cytometry should be referenced as well.
6) Do not forget Supplemental 1 legend.
Author Response
Reviewer 1 Comments
The authors have reinforced the manuscript as suggested in the first reviewing. The statistics data accompanied by the modification of the histogram strongly reinforce the manuscript and the conclusions. This shows the amount of work behind the data.
I still have a few comments that need to be modified before publications:
1) 'The CS patients seems to exhibit nearly the same pattern for the examined parameter as in elderly group (41-60years (Figure 1)". Now that we have a better overview, I don't think authors can conclude anything as it is not significant. I don't think it is necessary to conclude something here.
R1: We would like to thank the reviewer for his comment. We deleted the information as suggested. Line158-159.
2) "Results showed that the anti-inflammatory cytokines IL-4, IL-5, IL-10 and IL-13 slightly increase in CS patients compared to heathy donors". Authors should emphasize here that regarding the low number of patients, it is not statistically significant but there is a trend. Authors can also re-emphasize how difficult it is to obtain patients. This is completely fine.
R2: We would like to thank the reviewer for his suggestion. We have made the requested modification. Line 213-214.
3) "Lymphocyte T" should be written "T lymphocyte".
R3: We have made the requested modification. Line 27; Line 69.
4) Some images are blurry.
R4: We have improved the quality of the figures as requested.
5) References to all product used must be indicated in the protocol. Antibodies used for flow cytometry should be referenced as well.
R5: We have added the requested information line 122-123 and line 137-140.
6) Do not forget Supplemental 1 legend.
R6: We have added the appendix section which contains the requested legend of supplementary Figure1. Line 422-425.